# Incidence of Perinatal Post-Traumatic Stress Disorder in Catalonia: An Observational Study of Protective and Risk Factors

**DOI:** 10.3390/healthcare12080826

**Published:** 2024-04-13

**Authors:** Olga Canet-Vélez, Meritxell Escalé Besa, Montserrat Sanromà-Ortíz, Xavier Espada-Trespalacios, Ramón Escuriet, Blanca Prats-Viedma, Jesús Cobo, Júlia Ollé-Gonzalez, Emili Vela-Vallespín, Rocio Casañas

**Affiliations:** 1Global Health, Gender and Society (GHenderS), Blanquerna School of Health Sciences, Ramon Llull University, 08025 Barcelona, Spain; olgacv@blanquerna.url.edu (O.C.-V.); mescale@tauli.cat (M.E.B.); xespada@catsalut.cat (X.E.-T.); rescuriet@gencat.cat (R.E.); juliaog@blanquerna.url.edu (J.O.-G.); rociocs@blanquerna.url.edu (R.C.); 2Official College of Nurses of Barcelona, 08019 Barcelona, Spain; 3Blanquerna School of Health Sciences, Ramon Llull University, 08022 Barcelona, Spain; 4Catalan Health Institute, 08007 Barcelona, Spain; 5Department of Gynegology and Obstetrics, Parc Taulí University Hospital, 08208 Sabadell, Spain; 6Igualada Health Campus, University of Lleida, 25002 Igualada, Spain; 7Catalan Health Service (CatSalut), Catalan Department of Health, 08028 Barcelona, Spain; 8Public Health Agency of Catalonia (ASPCAT), Catalan Department of Health, 08005 Barcelona, Spain; blanca.prats@gencat.cat; 9Perinatal Mental Health Program, Mental Health Department, Parc Taulí University Hospital, 08208 Sabadell, Spain; jcobo@tauli.cat; 10Department of Psychiatry and Forensic Medicine, Autonomous University of Barcelona, 08193 Barcelona, Spain; 11Information Systems, CatSalut, 08028 Barcelona, Spain; evela@catsalut.cat; 12Digitalization for the Sustainability of the Healthcare System (DS3), IDIBELL, 08908 Barcelona, Spain

**Keywords:** post-traumatic stress disorder, PTSD, perinatal, risk factors, protective factors

## Abstract

Pregnancy and childbirth have a great impact on women’s lives; traumatic perinatal experiences can adversely affect mental health. The present study analyzes the incidence of perinatal post-traumatic stress disorder (PTSD) in Catalonia in 2021 from data obtained from the Registry of Morbidity and Use of Health Resources of Catalonia (MUSSCAT). The incidence of perinatal PTSD (1.87%) was lower than in comparable studies, suggesting underdiagnosis. Poisson regression adjusting for age, income, gestational weeks at delivery, type of delivery, and parity highlighted the influence of sociodemographics, and characteristics of the pregnancy and delivery on the risk of developing perinatal PTSD. These findings underline the need for further research on the risk factors identified and for the early detection and effective management of PTSD in the perinatal setting.

## 1. Introduction

Neurobiological and hormonal conditions present throughout pregnancy, during childbirth, and at the start of breastfeeding prepare women for imprinting and bonding with their newborn children [1,2]. Events occurring during childbirth and in the first hours thereafter can be deeply recorded in memory, whether they are positive or traumatic [3]. Thus, childbirth makes women vulnerable to developing maternal mental illness, among the most common of which are depression, anxiety, and post-traumatic stress disorder (PTSD) [4,5,6]. These mental illnesses can manifest during the perinatal period, which comprises the time from the beginning of pregnancy to one year after childbirth [7].

The fifth edition of the Diagnostic and Statistical Manual of Mental Disorders (DSM-5) defines PTSD as recurrent, disturbing memories of an unbearable traumatic event that begin within 6 months of the event and last for more than 1 month [8]. The DSM-5 groups the main symptoms of PTSD in four areas (persistent re-experiencing of the traumatic event in flashbacks and/or nightmares; avoidance of stimuli associated with the event; negative changes in cognition and mood; and physiological hyperactivation on exposure to stimuli related to the event). The diagnosis of PTSD requires to impact the individual’s functioning. PTSD can become chronic [8].

The scientific community first became interested in PTSD when it was noted that some Vietnam veterans had developed this anxiety disorder as a sequela after returning from the war, and it was included in DSM-III in 1980 [9]. DSM-IV broadened the definition of trauma, and childbirth was considered sufficiently traumatic to result in perinatal PTSD [4,10,11,12]. Any threat to the mother and/or newborn child during childbirth that causes the mother to experience extreme fear, helplessness, or horror can trigger perinatal PTSD [12,13,14]. Common symptoms of perinatal PTSD include opposition or refusal to become a mother, derealization, irritability, insomnia, and failure to bond with the newborn [13,14,15].

The reported prevalence of perinatal PTSD varies considerably. In a 2017 systematic review and meta-analysis of 35 studies of prenatal PTSD and 28 studies of postpartum PTSD, Yildiz et al. [6] found the prevalence ranged from 0% to 35% during pregnancy and from 0% to 21% in the year after childbirth. In another systematic review published in the same year, Geller and Stasko [16] found similar prevalences: from 2.3% to 24% during pregnancy and from 1% to 20% in the year after childbirth.

The wide variability in the prevalence of perinatal PTSD can be explained by the diversity of risk factors [5,6,17]. Factors associated with perinatal PTSD include the mother’s history of mental disease [18,19] and domestic violence or sexual violence before childbirth [18,20], obstetric violence [21,22,23], lack of emotional support from healthcare professionals or from family and friends [20,24], health problems in the newborn child [25], and sociodemographic characteristics such as financial problems, living in urban areas, and belonging to an ethnic minority [26,27]. In women with these risk factors, the reported prevalence of prenatal PTSD is as high as 18.95%, compared to 4% in the wider community sample from the same study [6]; another study found similar results, reporting 15.7% prevalence in women with risk factors, compared to an estimated 3.1% in the community population.

There are studies in Spain, that have focused on risk factors of perinatal PTSD. Hernández-Martínez et al. [28] found that risk for perinatal PTSD 4 to 6 weeks after childbirth was 10.6% [28]. In another study, the same group reported that the prevalence of PTSD in women at risk one year after childbirth was 7.2% [17]. A third study in which this group collected data online 1 to 3 years after childbirth found that 13.1% of women had a high risk of PTSD, defined as Perinatal Post-Traumatic Stress Disorder Questionnaire scores ≥19 [5].

Women who have experienced disrespect and abuse during childbirth have higher rates of postpartum depressive symptoms [29]. In a multicenter observational cohort study in 25 French maternity hospitals, mental health during the postpartum period and its relationship to disrespect during childbirth was assessed. Postpartum depression was present in 10.56% (*n* = 13/123) of mothers, and 4.06% (*n* = 5/123) could be considered as suffering from post-traumatic stress disorder related to childbirth. Disrespect was positively associated with postpartum depression at 2 months after delivery [30].

A not very studied factor that may be related to the onset of perinatal PTSD is severe fear of childbirth and the stress it can generate before delivery, called pre-traumatic stress. Negative anticipation of childbirth may help the development of PTSD [19].

The experience of having a newborn hospitalized in the Neonatal Intensive Care Unit (NICU) can be very traumatic for the caregivers of the infant, especially for mothers. Both preterm birth and NICU hospitalization generate some mental distress for mothers that can develop into perinatal PTSD [31].

Not only can the health of the newborn affect the mental health of the mother, developing PTSD, but also vice versa. One of the consequences of developing perinatal PTSD is the impairment of the mother’s bond with the newborn [32]. The early relationship between newborns and their caregivers is critical to the future mental health and well-being of infants. Symptoms of PTSD and depression can interfere with a mother’s ability to parent her children, thus affecting the bond [33]. Perinatal PTSD is linked to an avoidant attachment style associated with rejection and anger [33]. The development of a secure and organized infant attachment depends on caregiving and nurturing behaviors characterized by high sensitivity, responsiveness, and nurturance [34].

In addition, the symptoms of perinatal PTSD affect the relationships in the mother’s environment. In particular, the impact on the relationship with family and partners, generated from the impairment of the mother–baby bond, is highlighted [35].

PTSD is thought to be underdiagnosed and therefore undertreated; failure to diagnose perinatal PTSD suggests a lack of sufficient preventive and diagnostic strategies for this disorder [36,37]. For this reason, obstetric care providers should evaluate all pregnant women with a validated instrument to detect PTSD and have systems to guarantee early diagnosis and appropriate treatment [38]. Few questionnaires to assess perinatal PTSD have been validated [38]. One instrument specific for postpartum PTSD is the City Birth Trauma Scale (City-BiTS), a questionnaire consisting of 29 items developed in accordance with the DSM-5 criteria [39]; a version of the City-Bits adapted to the Spanish population has been validated [40].

Left untreated, maternal mental health disorders result in increased morbidity and mortality in both mothers and newborns, as well as increased healthcare costs [11]. Current treatments for PTSD include trauma-focused cognitive behavioral therapy, exposure therapies, stress inoculation therapy, and in some cases medication [41]. These treatments are, however, experimental and still under development. Prior studies recommend increasing medical and emotional support to prevent perinatal PTSD, analyzing and trying to improve the preparation for childbirth through prenatal classes and pain reduction strategies, and paying special attention to mothers with a history of mental health problems or sexual violence [42]. The evidence suggests that detecting maternal mental health problems alone can have positive effects [43] and that treatment can provide the greatest benefits and reduce the duration of PTSD [44].

For women at risk of developing symptoms of childbirth-related post-traumatic stress disorder, early psychological screening can be useful to initiate the necessary early psychological support [45]. Combined programs specifically targeting mood and attachment problems have shown to be successful [46]. Early interventions using traditional trauma-focused therapies, psychological counseling and those focusing on dyadic mother–child approach strategies have also shown a positive treatment effect [47].

Few studies have measured the incidence of perinatal PTSD in Spain, and scant data are available about this disorder in the region of Catalonia. Thus, further studies are necessary to know the true extent of perinatal PTSD and to better understand the factors related with it, so that women can receive the care that they need. Moreover, accurate estimations of the incidence of perinatal PTSD will enable health authorities to know the resources that need to be invested to prevent and treat it and the repercussions on public health. The current study aimed to estimate the incidence of perinatal PTSD in Catalonia to help determine how to improve care for the women who develop it and their newborn babies.

## 2. Materials and Methods

### 2.1. Design

This cross-sectional observational study analyzed data from the period comprising January 2021 through December 2021 retrieved from the Catalan Registry of Morbidity and Use of Healthcare Services (MUSSCAT), managed by the Catalan Health Service (CatSalut), ministry of health of the Catalan government (autonomous region of Spain).

### 2.2. Population

The study population comprised all women who gave birth in public hospitals in Catalonia in 2021. We excluded births in which the gestational week was not recorded, miscarriages, and abortions.

### 2.3. Sample Size

The sample comprised all 39,375 women who gave birth in public hospitals in Catalonia in 2021.

### 2.4. Recruitment

MUSSCAT compiles information about health and public healthcare services at primary care centers and hospitals for use in public health studies and healthcare management. Among other functions, the register integrates and homogenizes all information about primary and secondary diagnoses to the Minimum Basic Dataset reported by the different care providers (primary care, hospital care, emergency departments, specialized care, and both community and hospital mental health programs). We used MUSSCAT because is the Catalan Health Surveillance System, an administrative database within the Catalan Health Service (CatSalut), which is the main provider of healthcare assistance in Catalonia. Its automated data validation system checks the consistency of the data and identifies potential errors.

### 2.5. Variables and Measurements

Two sociodemographic variables were analyzed: age in years (grouped into <20, 20–24, 25–29, 30–34, 35–39, 40–44, or >44) and annual income in euros, stratified according to pharmaceutical copay groups: high (>EUR 100,000), medium (EUR 18,000–EUR 100,000), low (<EUR 18,000), or very low (dependent on public subsidies) [48].

Deliveries were classified into the following categories according to All Patient Refined Diagnosis Related Groups (APR DRG) v36 [49]: cesarean, delivery with sterilization, delivery with a surgical procedure other than dilation and curettage, and delivery. Cases of PTSD were identified through the International Classification of Diseases, Tenth Revision, Clinical Modification (ICD-10-CM) codes [50]. The diagnostic codes for PTSD were PTSD; PTSD, unspecified; PTSD, acute; and PTSD, chronic. A woman was considered to have PTSD when a diagnosis of the listed indicators was recorded in the period comprising the 9 months prior to childbirth through 12 months after childbirth.

Cases were classified according to the following variables related to delivery: degree of intervention (non-instrumental vaginal delivery, instrumental vaginal delivery, or cesarean delivery), the number of births (single or multiple), and the gestational week at delivery. We also collected information about the variable parity, understood as the number of previous births, and described as follows: “1st birth” refers to women who have not had a previous birth and this is their first, “2nd birth” refers to women who have had one previous birth and this is their second, “3rd birth” refers to women who have had two previous births and this is their third, and “>3 births” refers to women who have had more than three previous births.

### 2.6. Ethical Considerations

Our institutional ethics committee approved the study (CER-FCSB # 2023-07-01), waiving the requirement for informed consent because all data were anonymized in accordance with Spanish law (Organic Law 3/2018) and European Union directives (EU Regulation 2016/679). The study adhered to the principles of the Helsinki declaration.

### 2.7. Statistical Analysis

We report categorical variables as frequencies and percentages and continuous variables as means and standard deviations. Incidence rates are expressed as percentages, and to calculate significance, we used Pearson’s Chi-squared test.

To identify risk factors for perinatal PTSD, we used Poisson regression, adjusting for age, income, gestational week, type of delivery, and parity.

We used R, version R-4.2.0, for all analyses.

## 3. Results

A total of 39,375 women gave birth in public hospitals in Catalonia in 2021. Of these, 737 (1.9%) were diagnosed with perinatal PTSD. Table 1 reports the incidence of perinatal PTSD according to age and income. Statistically significant differences were observed in the diagnosis with reference to the age of the women. Those who had an age <20 years old had the highest rate (3.5%), followed by women between 40 and 44 years old (2.4%); women between 25 and 29 years old had the lowest rate (1.5%). Income also showed significant differences, whereby women with very low incomes had the highest rate of PTSD (2.3%), and women with low or medium incomes had similarly lower rates (1.8%). None of the women with high incomes were diagnosed with PTSD.

Table 2 reports the incidence of perinatal PTSD according to obstetric variables. Women who had cesarean sections had the highest incidence of PTSD (2%), followed by those who had instrumental vaginal births (1.9%) and those who had non-instrumental vaginal births (1.8%). The incidence of PTSD was higher in multiple pregnancies (2.3%) than in single pregnancies (1.9%). With statistically significant differences, it was observed that in women who gave birth prematurely (<37 gestational weeks), the incidence of PTSD (3.3%) was nearly twice the 1.8% rate observed in women with full-term pregnancies (37–42 gestational weeks).

In the multivariate analysis (Table 3), an age of 25 to 29 years or 30 to 34 years was a protective factor. The relative risk of perinatal PTSD was 1.4 times higher in women with very low incomes than in those with other income levels. The risk of PTSD was higher in women with instrumental vaginal births. Giving birth after at least 38 weeks of gestation had a protective effect against developing PTSD.

After adjusting for age, income, gestational weeks, and type of birth, the incidence rate for the entire population of Catalonia was 18.7 cases per 1000 births. Table 4 shows the adjusted rates for each health district in the region.

## 4. Discussion

This study aimed to determine the incidence of perinatal PTSD in women who gave birth in centers belonging to the Catalan Health Service and to analyze characteristics and factors that could influence the incidence. We found that 19 of every 1000 pregnant women attended in 2021 developed perinatal PTSD. The rate of perinatal PTSD was highest in the age groups where pregnancy is less common (20–24 years and 40–44 years), in women with very low incomes, in those who had instrumental vaginal births or cesarean sections, in those with multiple pregnancies, and in first-time mothers.

The incidence of perinatal PTSD in our study (1.87%) is much lower than that reported in other studies carried out in similar contexts. Studies in women in Spain reported about 10% of women were at risk of perinatal PTSD [5,17]. The low incidence rate in our study might be due to underdiagnosis, underlining the need to screen women in the perinatal period for PTSD with validated instruments such as the City Trauma Scale [39] to identify those with symptoms of PTSD in whom the syndrome has not been diagnosed.

Despite the difference in the rates among studies, the characteristics that influenced the probability of developing perinatal PTSD in our study were similar to those identified in other studies; the mother’s age and income, type of delivery, parity, and number of babies expected can increase or decrease the risk of perinatal PTSD.

In the present study, giving birth between the ages of 25 and 34 was associated with decreased risk of developing perinatal PTSD, and in some other age groups (20–24 years or 40–44 years), giving birth was associated with a greater risk of developing perinatal PTSD. This finding is in line with those reported elsewhere, where pregnancy at young ages increased the risk of developing mental health problems [51].

Dekel et al. concluded that the risk of perinatal PTSD varies with the type of birth and the mother’s experience during the delivery, with the probability of developing perinatal PTSD being higher when the delivery is painful, when the woman does not have control, or when there is obstetric violence. Dekel et al. showed that the risk of perinatal PTSD was of 14.7% for vaginal delivery and of 41.2% for unplanned cesarean section [52]. We also found that women who had cesarean sections or instrumental vaginal births had a greater risk of PTSD and that those who had non-instrumental vaginal births had fewer symptoms of PTSD. Along these lines, in a systematic review of studies including a total of 17,675 women, El Founti Khsim et al. [53] identified obstetric interventions and obstetric violence as factors associated with postpartum PTSD and accompaniment, information, and adherence to the mother’s birth plan as protective factors; however, this contribution is not conclusive.

We found that the incidence of perinatal PTSD was higher in women who gave birth prematurely (<37 weeks of gestation). Premature delivery is associated with more complications and risk for the newborn’s health, causing insecurity, fear, and anguish in the mother that can contribute to the development of perinatal PTSD, a study concluded that the prevalence of PTSD symptoms following preterm birth was 71.1% [54].

We found that the mother’s socioeconomic characteristics influence the appearance of the symptoms of perinatal PTSD. Women in the low- or very low-income brackets had a higher incidence of perinatal PTSD. These results corroborate those reported by Kastello et al. [55] in a cross-sectional study of 239 women with low incomes exposed to domestic violence in the USA. Arcaya et al. [56] point out that low income and educational level are associated with greater difficulties in accessing health services. This difficulty may help explain why women with low or very low socioeconomic levels have a greater tendency to develop perinatal PTSD. It stands to reason that having access to quality healthcare and having a favorable environment and support network, often associated with comfortable financial situations, would protect women against mental health disorders related to childbirth. In the present study, none of the women in the high-income group were diagnosed with perinatal PTSD. This finding could be related with the low number of women in this group, likely related to the preference for private health services in this income group. We also found that the health districts with the lowest incomes had the highest incidence rates of perinatal PTSD, with 2.79% in Metro South and 2.55% in Metro North, compared to 1.87% in all of Catalonia.

PTSD is a major mental health problem worldwide, and if a woman is exposed to trauma, it can be associated with risk of maternal and infant complications [57]. Strong maternal attachment is associated with the decrease in symptoms of depression or anxiety [58]. In contrast, maternal psychological distress, which is common in the perinatal period, is associated with adverse outcomes [59]. These clinical situations lead to an increased risk of postpartum mother–child bonding problems, as the biobehavioral synchrony that underpins attachment is altered [60]. There is evidence of the need for screening and follow-up of postpartum depressive symptoms, together with psychosocial stressors in women [61].

In summary, our analysis of perinatal PTSD in Catalonia shows that the following factors are associated with this syndrome: the type of delivery, pregnancy, and the mother’s socioeconomic level. Thus, we need to pay special attention to these women at risk, avoiding practices that increase the risk of developing perinatal PTSD and improving healthcare, especially in relation to the most vulnerable groups, and we need to investigate how best to accomplish these goals.

### Limitations

All our data came from the public health system, so including data from private centers may have yielded different results. Our data were extracted from the Health Department’s database, which did not include some variables identified in other studies that are associated with perinatal PTSD, such as a history of mental health problems [18,19], partner or sexual violence [18,20], or obstetric violence during delivery [21,22,23]. The absence of these data may give insufficient information of the main risk and protective factors for perinatal PTSD. Future studies should delve deeper into these questions through broader data collection with questionnaires that include items about known and potential risk factors and protective factors.

In addition, the data obtained from the Health Department’s database only reflect perinatal PTSD cases that have been formally diagnosed, which leaves out many people who may have experienced symptoms but have not been recognized as having the disorder. Previous studies support the idea that there is underdiagnosis of PTSD in the primary care setting [62]. Furthermore, the low incidence of perinatal PTSD in the present study (1.87%), in comparison to others, might show an underdiagnosis of the disorder. Studies which used perinatal PTSD validated questionnaires, reported that about 10% of women were at risk of developing perinatal PTSD [5,17].

The present study shows the importance of creating tools and protocols for a more accurate detection of PTSD. This will improve the early detection and treatment of PTSD, providing a more accurate picture of the real incidence in this context.

## 5. Conclusions

The low incidence of perinatal PTSD in Catalonia found in the current study might be related to underdiagnosis. Factors such as age, income level, district of residence, type of pregnancy, type of delivery, or gestational weeks at birth influence the likelihood of perinatal PTSD.

A predictive model to identify women at risk of developing perinatal PTSD would enable the provision of special care for women at risk, avoiding practices that contribute to the development of the disorder and improving healthcare.

## Figures and Tables

**Table 1 healthcare-12-00826-t001:** Incidence of perinatal PTSD in Catalonia broken down by sociodemographic variables.

Variable	Population (*n*)	Cases of Perinatal PTSD (*n*)	%	*p*-Value
**Age**				<0.01
<20	622	22	3.5%	
20–24	3634	77	2.1%	
25–29	8021	119	1.5%	
30–34	12,848	216	1.7%	
35–39	10,516	216	2.1%	
40–44	3427	82	2.4%	
>44	307	5	1.6%	
**Income**				0.035
High	62	0	0.0%	
Medium	10,228	181	1.8%	
Low	23,844	433	1.8%	
Very low	5241	123	2.3%	

**Table 2 healthcare-12-00826-t002:** Incidence of perinatal PTSD in Catalonia (2021), according to obstetric variables.

Variable	Population (*n*)	Cases of PTSD (*n*)	%	*p*-Value
**Gestational weeks**				<0.01
<37	2511	83	3.3%	
37	2943	77	2.6%	
38	5491	102	1.9%	
39	10,227	206	2.0%	
40	11,333	187	1.7%	
41	6459	78	1.2%	
>41	411	4	1.0%	
**Type of delivery**				0.5
Non-instrumental vaginal	25,655	467	1.8%	
Instrumental vaginal	4161	78	1.9%	
Cesarean	9559	192	2.0%	
**Type of pregnancy ***				0.5
Single	38,666	721	1.9%	
Multiple	709	16	2.3%	
**Parity ****				0.8
1st birth	22,305	428	1.9%	
2nd birth	12,083	219	1.8%	
3rd birth	3561	67	1.9%	
>3 births	1426	23	1.6%	

* According to the number of fetuses developed in the uterus (single pregnancy: 1 fetus; multiple pregnancy: ≥2 fetuses). ** Number of births, including the current one.

**Table 3 healthcare-12-00826-t003:** Multivariate analysis of sociodemographic and obstetric variables in the risk of perinatal PTSD. Catalonia 2021.

Variable	Unstandardized Beta Coefficient	RR *	95%CI **
**Age group**			
<20	0	1	-
20–24	−0.425	0.654	0.406–1.052
25–29	−0.738	0.478	0.302–0.758
30–34	−0.587	0.556	0.355–0.871
35–39	−0.379	0.684	0.435–1.076
40–44	−0.266	0.767	0.472–1.244
>44	−0.723	0.485	0.182–1.292
**Income**			
High	−11.282	0	0.000–0.000
Medium	0	1	-
Low	0.05	1.051	0.878–1.259
Very low	0.32	1.377	1.073–1.768
**Type of birth**			
Non-instrumental vaginal	0	1	-
Instrumental vaginal	0.066	1.068	0.836–1.364
Cesarean	−0.007	0.993	0.832–1.184
**Type of pregnancy *****			
Single	0	1	–
Multiple	−0.232	0.793	0.473–1.327
**Gestational weeks**			
<36	0	1	–
36	0.118	1.125	0.726–1.743
37	−0.199	0.820	0.572–1.175
38	−0.542	0.581	0.411–0.821
39	−0.455	0.635	0.461–0.873
40	−0.657	0.518	0.375–0.717
41	−0.960	0.383	0.266–0.552
>41	−1.182	0.307	0.110–0.852

* RR: relative risk; ** CI: confidence interval; *** according to the number of fetuses developed in the uterus (single pregnancy: 1 fetus; multiple pregnancy: ≥2 fetuses).

**Table 4 healthcare-12-00826-t004:** Adjusted incidence rate (*) according to health districts. Catalonia 2021.

Health District	Population (n°)	Cases of PTSD (n°)	Incidence Rate (×1000 Births)	** 95CI% of the Rate
All Catalonia	39,375	737	18.7	--
Lleida	2251	23	10	6.6–15.0
Tarragona	3882	26	6.6	4.5–9.7
Ebro region	1024	7	6.6	3.1–13.9
Girona	5625	72	12.7	10.1–16.0
Central Catalonia	3097	55	18.2	14.0–23.7
Pyrenees	313	5	15.9	6.6–38.3
Metro South	6291	175	27.9	24.0–32.3
Metro North	9712	246	25.5	22.5–28.9
Barcelona city	7180	128	17.9	15.1–21.3

* Adjusted for age, income, gestational weeks, and type of birth. ** Confidence interval.

## Data Availability

All data used in this study were obtained from the Registry of Morbidity and Use of Health Resources of Catalonia (MUSSCAT), maintained by the Catalan Health Service (CatSalut). The availability of these data is subject to the policies and regulations established by CatSalut (https://catsalut.gencat.cat/ca/inici/index.html) accessed on 20 July 2023.

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
