# Peer review of "Incidence of Perinatal Post-Traumatic Stress Disorder in Catalonia: An Observational Study of Protective and Risk Factors"

_healthcare, 2024, doi:10.3390/healthcare12080826_

Round 1

Reviewer 1 Report

Comments and Suggestions for Authors

Dear authors,

It has been a pleasure for me to review this manuscript that attempts to analyze the incidence of perinatal post-traumatic stress disorder in Catalonia.

I found the topic interesting and novel since there is not much research on perinatal post-traumatic stress disorder. Furthermore, it seems to me that the manuscript is very well structured and follows an appropriate methodology.

The introduction section seemed correct to me, since it immediately defines the problem and uses updated evidence to place us in the context of the research. At the end of this section the objective of the research is clearly defined.

The materials and methods section also seems correctly developed to me, but I would like to make a suggestion. Between lines 121 and 124 the ethical considerations of the study are detailed. I think this information should not be within section 2.2 population. I suggest creating a new section 2.6 called ethical considerations. In this section you can put that information and even detail it a little more.

The results section also seemed correct to me, I think the results were presented clearly.

I also found the discussion section well developed. The results of the study were discussed and in the final part the authors explained the limitations of this study.

I think the conclusions are well written and faithfully support the results obtained in the study.

I think this manuscript is very interesting to be published.

Kind regards

Author Response

Dear Reviewer,

We are very grateful for the comments you have given us. Please find the detailed responses below and the corresponding corrections highlighted in the re-submitted files:

Comment 1: The materials and methods section also seems correctly developed to me, but I would like to make a suggestion. Between lines 121 and 124 the ethical considerations of the study are detailed. I think this information should not be within section 2.2 population. I suggest creating a new section 2.6 called ethical considerations. In this section you can put that information and even detail it a little more

Response 1: We agree with this comment. We have made the change you proposed to us, on moving the content of lines 121 to 124 to a new section: 2.6 Ethical considerations. (Page 4. Lines 191-194)

Thank you very much for the review.

Reviewer 2 Report

Comments and Suggestions for Authors

The article titled “Incidence of Perinatal Posttraumatic Stress Disorder in Catalonia: an Analysis of Protective and Risk Factors" represents an interesting article on incidence of perinatal post-traumatic stress disorder (PTSD) in Catalonia. I am reporting some comments as follows in order to recommend some implementations.

For the authors

Title:

The title could be improved. It highlights "the analysis of risk and protective factors," instead, it would be appropriate to highlight the observational nature of the study and the source of the data objects of the study (Registry of Morbidity and Use of Health Resources of Catalonia) already in the title.

Introduction:

The authors should implement the introduction of the article with more references to maternal mental health, highlighting the relationship between stress in pregnancy, perinatal depression, and other psychological variables that influence the onset of PTSD.

Authors could implement the manuscript by including other references to international scientific literature, as long as they are of recent publication.

Discussion:

In addition, to highlight the clinical implications of their study, it is recommended to expand the discussion on the impact of maternal health on child development.

Authors could implement the manuscript by including other references to international scientific literature, as long as they are of recent publication.

Limitation session:

The authors should better discuss the limitations of how diagnosis data are retrieved. From the description, MUSSCAT appears to contain information on all types of diagnoses recognized and attributed to patients during their access to health services. However, this results in a huge gap with respect to all cases that have not been diagnosed and recognized. It is worth devoting even more space to discussing this in the limitations section. At the same time, another major limitation of this type of collection is the inability to consider many other psychological variables that can play the role of a risk or protective factor for PTSD; this, too, needs to be discussed more fully in the limitations.

Author Response

Dear Reviewer,

We are very grateful for the comments you have given us. Please find the detailed responses below and the corresponding corrections highlighted in the re-submitted files:

Comment 1: Title:

The title could be improved. It highlights "the analysis of risk and protective factors," instead, it would be appropriate to highlight the observational nature of the study and the source of the data objects of the study (Registry of Morbidity and Use of Health Resources of Catalonia) already in the title.

Response 1: We agree that highlighting the observational nature of the study is crucial for a more complete understanding of the research.

We have taken your suggestion into consideration and have updated the title of our study to more accurately reflect its observational nature. However, we decided not to include the source of the data in the title for reasons of brevity and clarity. We believe that by mentioning it in the body of the paper, we provide the necessary information about the provenance of the data without overloading the title.

The new proposed title is "Incidence of Perinatal Posttraumatic Stress Disorder in Catalonia: an Observational study of Protective and Risk Factors".

Comment 2: Introduction:

The authors should implement the introduction of the article with more references to maternal mental health, highlighting the relationship between stress in pregnancy, perinatal depression, and other psychological variables that influence the onset of PTSD.

Authors could implement the manuscript by including other references to international scientific literature, as long as they are of recent publication.

Response 2:

We have incorporated updated references and the required topics (maternal mental health, highlighting the relationship between stress in pregnancy, perinatal depression and other psychological variables), we are grateful for your comments, as they have improved the article. (Page 1. Lines 38-40) (Page 2. Lines 83-92)

Comment 3: Discussion:

In addition, to highlight the clinical implications of their study, it is recommended to expand the discussion on the impact of maternal health on child development.

Authors could implement the manuscript by including other references to international scientific literature, as long as they are of recent publication.

Response 3:

We thought it would be useful to incorporate information about the impact of maternal mental health on newborn development in the discussion. We also incorporate information about these aspects in the introduction. (Pages 2-3. Lines 93-108) (Page 8. Lines 303-311)

Comment 4: Limitation session:

The authors should better discuss the limitations of how diagnosis data are retrieved. From the description, MUSSCAT appears to contain information on all types of diagnoses recognized and attributed to patients during their access to health services. However, this results in a huge gap with respect to all cases that have not been diagnosed and recognized. It is worth devoting even more space to discussing this in the limitations section. At the same time, another major limitation of this type of collection is the inability to consider many other psychological variables that can play the role of a risk or protective factor for PTSD; this, too, needs to be discussed more fully in the limitations.

Response 4:

We agree with this comment. We have incorporated your recommendation to expand the limitation section. We have extended the discussion on the underdiagnosis of perinatal PTSD, as well as the absence of data on possible protective or risk factors. We recognise the importance of addressing these limitations for a more complete understanding of the research and to inform future research in the field. (Page 9. Lines 324-339).

Thank you very much for the review.

Reviewer 3 Report

Comments and Suggestions for Authors

No statistical significance test regarding the reported decrease in incidence of perinatal PTSD.

Author Response

Dear Reviewer,

We are very grateful for the comments you have given us. Please find the detailed responses below and the corresponding corrections highlighted in the re-submitted files:

Comment 1: No statistical significance test regarding the reported decrease in incidence of perinatal PTSD.

Response 1: We calculated the statistical significance of the incidence of PTSD using the Pearson's Chi-squared test, and the result has been added in a column in Tables 1 and 2 of the manuscript.

We have also added changes in the wording of the statistical analysis to include the statistical significance calculation test (Page 4. Line 198).

We have added to the results section references to the statistical significance of the studied variables (Page 5. Lines 205-206, 208-209, 219-220).

Thank you very much for the review.

Reviewer 4 Report

Comments and Suggestions for Authors

I read the paper very carefully and found the theoretical framework very clear.

The experimental design is simple and the statistical analysis method adequate.

The results seem to support the initial research hypotheses. Some doubts remain about the sample of subjects, the variables considered, and the mismatch of risk factors considered and those considered in previous
work in the literature.

I hope the authors will gladly accept my advice since I am proposing a major revision of their work: revision understood in these terms could make the work worthy of publication.

Line 130: Could you kindly specify why you used MUSSCAT?
Line 139: "Variables and measures" I think it is necessary to introduce other variables that describe the sample.

- Number of previous pregnancies: it is not clear whether the patients are first-time pregnancies or not.
- It would be interesting to understand whether a percentage of pregnant women have applied for psychological support in the past.

If possible, could you provide the results of the statistical analysis?
Line 221: It would be interesting to report the results in the literature review papers and compare the incidence of risk factors.

Author Response

Dear Reviewer,

We are very grateful for the comments you have given us. Please find the detailed responses below and the corresponding corrections highlighted in the re-submitted files:

Comment 1: Line 130: Could you kindly specify why you used MUSSCAT?

Response 1: We used MUSSCAT because is the Catalan Health Surveillance System, an administrative database within the Catalan Health Service (CatSalut), which is the main provider of healthcare assistance in Catalonia. The MUSSCAT includes information on morbidity and healthcare-related services within the public system, from assistance in hospitals or primary care to drug dispensation expenditure and it is a source of information for public health and healthcare management studies. MUSSCAT includes all the diagnosis reported by the different providers, regardless of whether they were recorded as the primary or secondary diagnosis. This information system collects all information from the entire public health system, including all hospital admissions and healthcare visits. Its automated data validation system checks the consistency of the data and identifies potential errors.

We modified line 130 of the manuscript: “We used MUSSCAT because is the Catalan Health Surveillance System, an administrative database within the Catalan Health Service (CatSalut), which is the main provider of healthcare assistance in Catalonia. Its automated data validation system checks the consistency of the data and identifies potential errors.” (Page 4. Lines 184-189).

Comment 2: Line 139: "Variables and measures" I think it is necessary to introduce other variables that describe the sample.

Response 2: Thank you very much for your suggestion, but being a retrospective observational study, we cannot add more variables than those used in the study design. The sociodemographic variables considered were age and income level, and the obstetric variables included were gestational age, parity, singleton or multiple pregnancy, and type of delivery. At the time of designing the study, it was not deemed appropriate to add more variables to describe the sample, but we will consider your suggestion when designing new observational studies to analyze incidence or prevalence.

Comment 3: Number of previous pregnancies: it is not clear whether the patients are first-time pregnancies or not.

Response 3: In the article, we do not discuss previous pregnancies; rather, we focus on parity (the number of times a woman has given birth). To describe the sample and analyse whether previous births influence the incidence of post-traumatic stress syndrome diagnosis, we use the variable parity. In this variable, "1st birth" refers to women who have not had a previous birth and this is their first, "2nd birth" refers to women who have had one previous birth and this is their second, "3rd birth" refers to women who have had two previous births and this is their third, and "> 3 births" refers to women who have had more than three previous births.

We have modified the description of the variable on line 149: "We also collected information about the mother’s past obstetric history" and added the variable parity, understood as the number of previous births, and described as follows: "1st birth" refers to women who have not had a previous birth and this is their first, "2nd birth" refers to women who have had one previous birth and this is their second, "3rd birth" refers to women who have had two previous births and this is their third, and "> 3 births" refers to women who have had more than three previous births. (Page 4. Lines 184-189).

Comment 4: It would be interesting to understand whether a percentage of pregnant women have applied for psychological support in the past.

Response 4: We appreciate your suggestion and agree with you. However, we have not collected data regarding previous requests for psychological support. We will take your suggestion into account for future analyses.

Comment 5: If possible, could you provide the results of the statistical analysis?

Response 5: We will forward your question to the research group, and to the extent possible, we will share the results of the statistical analysis.

Comment 6: Line 221: It would be interesting to report the results in the literature review papers and compare the incidence of risk factors.

Response 6: We agree with this comment. It is interesting to add information about the results of some previous studies on incidence according to risk factor. In this case, line 221 was intended to be an introduction to comment the risk and protective factors identified. For this reason, we have added the most notable findings results, below, throughout the discussion.

Page 8. Line 273-274: Dekel et al. shows that the risk of perinatal PTSD is 14,7% for vaginal delivery and 41,2% for unplanned caesarean section [52].

Page 8. Line 285-286: a study concluded that the prevalence of PTSD symptoms following preterm birth was 71,1% [54].

Thank you very much for the review.

Round 2

Reviewer 3 Report

Comments and Suggestions for Authors

I am happy with the revisions made

Reviewer 4 Report

Comments and Suggestions for Authors

well done !!!!